# X-ray Irradiation Reduces Live *Aspergillus flavus* Viability but Not Aflatoxin B1 in Naturally Contaminated Maize

**DOI:** 10.3390/toxins16080329

**Published:** 2024-07-25

**Authors:** Hannah Glesener, Darya Abdollahzadeh, Christopher Muse, Rosa Krajmalnik-Brown, Mark A. Weaver, Lee E. Voth-Gaeddert

**Affiliations:** 1Biodesign Center for Health Through Microbiomes, Arizona State University, Tempe, AZ 85281, USA; glesener@asu.edu (H.G.); dabdoll1@asu.edu (D.A.); christopher.l.muse@asu.edu (C.M.); dr.rosy@asu.edu (R.K.-B.); 2School for Engineering of Matter, Transport, and Energy, Arizona State University, Tempe, AZ 85281, USA; 3School of Sustainable Engineering and the Built Environment, Arizona State University, Tempe, AZ 85281, USA; 4USDA-ARS National Biological Control Laboratory, Stoneville, MS 38776, USA; mark.weaver@usda.gov; 5Center for Indigenous Health Research, Wuqu’ Kawoq|Maya Health Alliance, Chimaltenango 04006, Guatemala

**Keywords:** corn, aflatoxin, irradiation, *A. flavus*, food crops, laboratory safety

## Abstract

Food crops around the world are commonly contaminated with *Aspergillus flavus*, which can produce the carcinogenic mycotoxin aflatoxin B1 (AFB1). The objective of this study is to test an X-ray irradiation sterilization method for studying AFB1 in contaminated maize samples in the laboratory. Maize that had been naturally contaminated with 300 ppb AFB1 by the growth of aflatoxigenic *A. flavus* was ground and then irradiated at 0.0, 1.0, 1.5, 2.0, 2.5, and 3.0 kGy. *A. flavus* was quantified by dilution plating on potato dextrose agar (PDA) and modified Rose Bengal media (MDRB) for viability and qPCR for gene presence. AFB1 was quantified by HPLC and ELISA. *A. flavus* viability, but not gene copies, significantly decreased with increasing doses of radiation (PDA: *p* < 0.001; MDRB: *p* < 0.001; qPCR: *p* = 0.026). AFB1 concentration did not significantly change with increasing doses of radiation (HPLC: *p* = 0.153; ELISA: *p* = 0.567). Our results imply that X-ray irradiation is an effective means of reducing viable *A. flavus* without affecting AFB1 concentrations. Reducing the hazard of fungal spores and halting AFB1 production at the targeted dose are important steps to safely and reproducibly move forward research on the global mycotoxin challenge.

## 1. Introduction

Fungal toxins (mycotoxins) are a burden on the global food supply. In high-income countries, the burden is mainly economic due to crop loss [1]. In low- and middle-income countries, the burden includes poor health attributed to the consumption of contaminated food [1,2]. Research to understand the effects of chronic exposure on human health is ongoing [3,4]. Meanwhile, the global spread of mycotoxins in food crops is increasing and becoming increasingly unpredictable due to climate change conditions (e.g., water stress, elevated CO_2_, and temperature) [5]. When mycotoxins were first discovered in the 1960s, it quickly became clear that full eradication was not possible and humans would have to manage exposure through engineering controls in food processing systems [6]. The control points can generally be categorized as pre-harvest (e.g., protecting crops in the field) or post-harvest (e.g., proper storage of grains). Despite sophisticated technologies and successful interventions at both control points, mycotoxins are still pervasive in stored crops, particularly in communities that do not have the economic resources to implement industrial prevention systems [4]. It is paramount for scientists to innovate and implement solutions applicable across all control points within the food system.

Maize—the most abundantly grown crop by volume in the world and the third most consumed crop—is commonly contaminated by the carcinogenic aflatoxin B1 (AFB1) produced by *Aspergillus flavus* [7,8]. Scientists study maize in research labs to understand the complex relationships between food and fungi, yet there can be a disconnect between the characteristics of maize after lab processing compared to maize in the field. For example, sterilized maize can be artificially contaminated with AFB1 by spiking toxic doses into maize in the lab, yet this may not fully capture the critical morphological and biochemical processes that occur between the maize, fungus, and toxin [9]. While important insights about maize have been discovered through artificial contamination of maize with AFB1, it is important to study naturally occurring toxin formation by fungi for translational applications. However, natural contamination can introduce hazards to a lab environment [10,11].

When maize is brought into the lab, it can be sterilized to prevent the growth of microbes and to limit hazards from toxins in the lab. Sterilization is normally either physical (e.g., steam) or chemical (e.g., ethylene oxide) [12,13]. Both physical and chemical sterilization techniques affect the structure of the food product, interfering with certain applied science experiments such as nixtamalization [14]. An alternative sterilization technique is irradiation, a cold pasteurization technique used in industrial food supply systems because it limits structural changes to food, does not use heat, and effectively limits microbial growth. However, this technique is rarely used for research purposes in labs studying mycotoxin-contaminated food.

Researchers have mostly studied gamma irradiation using the radioactive decay of ^60^Co or ^137^Cs. Gamma irradiation is slowly being phased out because it is dangerous to operate, it is difficult to dispose of used materials, and the dose uniformity is highly variable. Electron beam (e-beam) irradiation is another option; however, this technique does not have good depth penetration and can mainly reduce surface contamination of foods. The third option is X-ray irradiation, which is estimated to be widely used in the future for food safety [15]. X-ray irradiation can be adapted on a smaller scale in the food industry without complex operating facilities, which means that irradiation units can be directly incorporated into continuous local food processes, saving the transportation and handling costs associated with larger, remote facilities used for irradiation [16]. Companies like Rayfresh Foods are working towards building this technology, though adoption is a hurdle due to the cost of equipment implementation and maintenance. Electricity costs and usage are also high, which is a downside to X-ray irradiation, yet the market for smaller, safer, and easily operated irradiation technology is increasing at the same time as research into the improved efficiency of renewable electricity [17]. Therefore, we chose X-ray irradiation as the cold-pasteurization method for this study [15].

Previous research on the irradiation of *A. flavus* and AFB1 on food products has mostly focused on reducing the fungal load for storage or decreasing the AFB1 content. Gamma irradiation (^60^Co) of ground pistachio nuts at increasing doses (0.0, 0.5, 1.0, 1.5, 2.0, 4.0, and 6.0 kGy) showed decreased *A. flavus* viability, with a 99.99% reduction at 4 and 6 kGy [18]. AFB1 degradation was observed in corn at increasing doses (4.0, 6.0, and 10.0 kGy) and found to have no significant reduction at 4 kGy but 72.11% and 84.23% at the respective higher doses [19]. Another study looking at *A. flavus* growth during grain storage showed that gamma irradiation (^60^Co) of whole maize (0, 2.5, 3.5, 4.5, and 6.0 kGy) decreased the viability of *A. flavus* at a proportional rate, with no growth seen at 6.0 kGy over 8 weeks [20]. X-ray irradiation (Rad Source 420M) of cannabis flowers (2.0, 2.5, and 5.0 kGy) showed no growth on potato dextrose agar (PDA) at or above 2.5 kGy [21]. It is unknown whether the X-ray irradiation results seen in cannabis flowers will translate to contaminated maize because of the different material matrix, though *A. flavus* reduction in maize gamma irradiation studies point to the likelihood that X-ray irradiation will also reduce *A. flavus* viability in maize.

This study tests the feasibility of X-ray irradiation as a sterilization technique for a lab-based, aflatoxigenic *A. flavus*-contaminated maize study. The objective is to test a sterilization method for studying toxins in food products in the laboratory using AFB1 produced by *A. flavus* on maize. The aim of this research is to understand the irradiation dose necessary to kill *A. flavus* on the maize while maintaining AFB1 concentrations for subsequent detoxification studies.

## 2. Results

### 2.1. X-ray Irradiation Decreases Aspergillus flavus Viability

Maize contaminated with AFB1-producing *A. flavus* was X-ray irradiated at increasing doses—0.0, 1.0, 1.5, 2.0, 2.5, and 3.0 kGy. As seen in Figure 1, dilution plating of irradiated samples on a modified Rose Bengal medium (MDRB; designed for *A. flavus*) shows no observed growth at or above 2.5 kGy. There was a consistent decline in observed colony-forming units (CFU) in samples as the irradiation dose increased.

For samples plated on MDRB, CFU per gram of maize decreased from an average of four and a half observed CFU at 0.0 kGy to no observed CFU at 2.5 and 3.0 kGy (Figure 2a). Dilution plating of irradiated samples on potato dextrose agar (PDA; designed for general fungi) showed similar results with no observed CFU at 3.0 kGy (Figure 2b). Each plate that contained at least one CFU was included in the log CFU calculations (Appendix A). Some treatments contained multiple plates with observed colonies (five observations for PDA irradiated at 1.0 kGy), while others only contained one plate with one observed colony (one CFU was observed on one plate at 2.0 kGy for MDRB and 2.5 kGy for PDA).

DNA extracts from the irradiated samples were amplified via a commercial qPCR kit, targeting the single-copy *A. flavus* MDR1 gene. qPCR showed consistent *A. flavus* presence across doses, with no detectable pattern of higher or lower concentrations between irradiation doses. The maize samples had a mean of 7.04 ± 0.34 log gene copies of MDR1 (Figure 3).

### 2.2. X-ray Irradiation Did Not Decrease Aflatoxin B1 Concentration in Maize

The irradiation dosage did not significantly reduce the AFB1 concentration as measured by HPLC (ANOVA: *p* = 0.153; Figure 4a). The mean AFB1 concentration of samples on the HPLC was 240.22 ± 17.66 ppb. The irradiation dosage did not significantly reduce the AFB1 concentration as measured by ELISA (ANOVA: *p* = 0.566; Figure 4b). ELISA tests showed a mean AFB1 concentration of 267.08 ± 53.59 ppb.

## 3. Discussion

We tested the feasibility of X-ray irradiation for sterilization of aflatoxigenic *Aspergillus flavus*-contaminated maize for laboratory-based aflatoxin B1 (AFB1) degradation work. We found that viable *A. flavus* was not detected in ground maize samples at an X-ray irradiation dose of 3.0 kGy. AFB1 concentrations were not significantly changed at a radiation dose up to 3.0 kGy. These findings build upon previous research showing that gamma irradiation and electron beam irradiation can be effective means of reducing the fungal concentration in contaminated foods [22,23]. It also builds on previous research demonstrating that AFB1 degradation did not occur below 4.0 kGy when treated with irradiation [19]. This is important for researchers who intend to study AFB1 in naturally contaminated samples while maintaining the integrity of the structure (physical/chemical properties) of the food product through cold pasteurization.

Irradiation acts in two primary ways to kill organisms—directly or indirectly. Indirect death occurs when high-energy particles turn cellular water into hydrogen peroxide and other cytotoxic radicals that can lead to cell death [24]. Direct death occurs when an organism’s reproductive DNA is destroyed. Because the MDR1 gene—the qPCR target gene in our experiment—is about 4.6 kbp long and the *A. flavus* genome is about 36.5 Mbp long, it is likely that a large number of copies of the target gene remained intact even if the genome was disrupted through irradiation. This could be a reason why qPCR results show that MDR1 gene copies remained high with increasing irradiation doses despite a reduction in viability. In addition, our results showed a more than two-log difference between fungal concentrations in plated samples compared to qPCR samples before radiation treatment. This could be due to dormant or dead *A. flavus* spores that are picked up by qPCR but not represented through plating. Previous research has shown that qPCR leads to higher counts than dilution plating, although both techniques are common and complementary practices in fungal microbiology [25].

Some limitations of this research are that we did not test any nutritional or physical characteristics of the maize product, which means we cannot draw conclusions about the efficacy of X-ray irradiation on food quality. However, the U.S. Food and Drug Administration (FDA) has approved irradiation for treatment of sprouting seeds up to 8.0 kGy, and our treatment was far below this, so it is likely that the organoleptic (sensory appeal) and nutritional quality of the maize remained acceptable [26]. Another limitation of this study is that we did not use whole maize kernels because we were restricted in size due to the size of the irradiator (15 g samples). The size limitation led to an uneven distribution of AFB1 in the maize, so ground maize was necessary to homogenize the contamination concentration. Whole maize kernels contaminated with AFB1 should be tested for food safety studies to ensure irradiation penetration; however, for the purpose of this study, ground maize sufficed.

These findings are important because they show the effective sterilization of a food product using non-radioactive, non-thermal, and non-chemical X-ray irradiation, which has rarely been performed on food and can be tested more broadly using other food matrices. Because mycotoxins are prevalent in the global food supply, it is critical that scientists study food products in the laboratory while keeping exposure to the hazards of mycotoxin-contaminated materials to a minimum. X-ray irradiation can help neutralize two out of three of the following primary hazards of working with AFB1-contaminated materials: inhalation of spores, contamination of other research materials, and toxic/carcinogenic quality. The carcinogenic hazard of AFB1 remains present and can be mitigated by proper handling. X-ray irradiation could be a good alternative to current sterilization techniques, especially given the safety profile of the technique and the expected increase in adoption and availability of equipment in the future. 

## 4. Materials and Methods

### 4.1. Chemicals and Media

The following chemicals were used for the extraction, purification, and quantification of aflatoxin B1 (AFB1) and *Aspergillus flavus.* All the following chemicals were purchased from Sigma-Aldrich (St. Louis, MO, USA): methanol (≥99.9%), acetonitrile (≥99.9%), acetone (≥99.5%), sodium chloride (≥99.0%, CAS: 7647-14-5), peptone (CAS: 91079-38-8), yeast extract (CAS: 8013-01-2), magnesium sulfate heptahydrate (≥99.0%, CAS: 10034-99-8), potassium phosphate monobasic (≥99.0%, CAS: 7778-77-0), Rose Bengal (≥80%, CAS: 632-69-9), streptomycin sulfate salt (CAS: 3810-74-0), chlortetracycline hydrochloride (≥75%, CAS: 64-72-2), 2, 6-dichloro-4-nitroaniline (96%, CAS: 99-30-9), agar powder (CAS: 9002-18-0), triton-X 100, potato dextrose agar (PDA) and aflatoxin mix (certified reference material: CRM46304, 1000 ug/kg AFB1). Ethanol (99.5%) was purchased from Decon Labs (Prussia, PA, USA). Glucose (≥99.5%, CAS: 50-99-7) was purchased from HiMedia Laboratories (Mumbai, Maharashtra, India). The modified Rose Bengal media (MDRB) contained (per liter) 30 g of sodium chloride, 20 g of agar, 10 g of glucose, 2.5 g of peptone, 0.5 g of yeast extract, 1 g of potassium phosphate monobasic, 0.5 g of magnesium sulfate heptahydrate, 25 mg of Rose Bengal (stored at 4 °C in 16.5 mg/mL ethanol), 4 mg of 2, 6-dichloro-4-nitroaniline (stored at 4 °C in 0.2 g/100 mL acetone), 187.5 mg of streptomycin (stored at 4 °C in 3.75 g/200 mL of 0.2 µm filtered water), and 15 mg of chlortetracycline (stored at 4 °C in 1.5 g/100 mL methanol). The solution was adjusted to pH 5.0 before adding agar, and the media was cooled to 60 °C before adding antibiotics [27].

### 4.2. Maize Inoculation and Sample Preparation

A composite grain sample was sourced from Guatemala in 2023 [2]. Intact white maize kernels were dried and stored at 4 °C before use in the present experiments. Maize was heated at 90 °C for 30 min to attenuate some of the microflora but was not sterilized to minimize chemical or structural changes. A portion of the grain sample was aliquoted as a negative control for the X-ray irradiation experiment, while the rest of this maize was moistened and inoculated in a biosafety cabinet with a mix of spores of aflatoxigenic *A. flavus* from Guatemala and representing common genotypes, as verified by DNA microsatellite analysis [2]. The maize was incubated, with samples collected and analyzed for AFB1 every 8–12 h, until the target concentration of 300 ppb AFB1 was reached. Contaminated grain was then dried, milled, and stored at 4 °C with minimal exposure to light.

### 4.3. X-ray Irradiation of Maize

Aflatoxin-contaminated ground maize was prepared for irradiation by dispensing 16 g in 58 × 70 mm plastic bags (2 mil). A subsample of six bags was irradiated by an X-ray using a 160KV RS2000 Series Biological Irradiator (Rad Source Technologies Inc., Buford, GA, USA) at level 5 (Figure 5). The equipment was operated for 35 cycles of 25 min of run time with a 30-minute rest period to preserve the X-ray bulb. Bags were removed from the irradiator at the target levels of 0.0, 1.0, 1.5, 2.0, 2.5, and 3.0 kGy occurring at cycles 0, 10, 15, 20, 25, 30, and 35. Immediately upon removal, the irradiated bags were labeled and stored at 4 °C.

### 4.4. Sample Aliquoting

X-ray-irradiated samples were aliquoted in a Labconco biosafety cabinet (Kansas City, MO, USA). Each bag of ground maize was emptied into a 50 mL Falcon tube and vortexed for 30 sec to homogenize the sample. Each 50 mL tube was then aliquoted for the following assays: 250 mg ± 10 mg directly into bead beating tubes for DNA extractions, 2 g ± 20 mg into a 10 mL tube for dilution plating, and 5 g ± 20 mg into a new 50 mL falcon tube for aflatoxin extraction. Experimental controls were also aliquoted in the biosafety cabinet in the same portions as experimental samples. The negative control was a portion of the composite grain sample before *A. flavus* inoculation. The positive control was a certified reference material with a known AFB1 concentration (200 ppb) purchased from Trilogy Analytical Laboratory (Washington, MO, USA). The samples were then stored at 4 °C in the dark until further analysis.

### 4.5. Aflatoxin B1 Extraction

Aflatoxin B1 (AFB1) was extracted from the maize samples based on previously published methods based on AOAC official method 993.16 [17]. In a chemical fume hood, 25 mL of 70% methanol was added with a serological pipette to each 5 g maize sample in a 50 mL conical vial. Each sample was vigorously shaken for 3 min, followed by centrifugation for 10 min at 4000 rpm. The liquid phase was filtered with a #1 Whatman filter. Each sample yielded between 12 and 15 mL of filtrate.

### 4.6. Aflatoxin B1 Quantification

The sample filtrate was aliquoted for ELISA (10 mL) and HPLC (2 mL) AFB1 analysis. ELISA samples were quantified with the AgraQuant^®^ Aflatoxin B1 ELISA kit (Romer Labs, Tulln, Austria). The kit range of quantification was 2–50 ppb, with a limit of detection (LOD) and limit of quantification (LOQ) of 2 ppb. Each maize filtrate was diluted twice to ensure AFB1 quantification, given the small range of AFB1 detection. Briefly, 10 mL of each sample filtrate was diluted with 40 mL of 70% methanol. Half of the resulting filtrate (25 mL) was further diluted with a 1:1 ratio of 25 mL 70% methanol (1:9 ratio of the original filtrate). The two diluted sample filtrates (100 µL of 1:4 and 1:9) were used to quantify AFB1 following the AgraQuant kit instructions. ELISA 96-well plates were read on a BioTek Synergy HTX Multimode Reader (Agilent Technologies, Santa Clara, CA, USA) with the monochromator set at 450 nm. Using the kit-provided AFB1 solutions, a five-point standard curve (0, 2, 5, 20, and 50 ppb) with triplicate readings had an R^2^ value of 0.9908.

For HPLC, each 2 mL sample was centrifuged at 10,000 rpm for 10 min, then filtered with a 0.2 µm PTFE filter into a 2 mL amber glass vial. Closely following Agilent Technologies technical application method, samples were injected (20 µL) and ran for 8 min in triplicate on an Agilent 1260 Infinity II HPLC using an Agilent Eclipse XDB-C18 4.6 × 150, 3.5 µm Rapid Resolution column (Agilent Technologies, Santa Clara, CA, USA) and photo diode array (PDA) detector set at a wavelength of 365 nm with a reference wavelength of 380 nm to observe AFB1 [27]. The mobile phase consisted of 50:40:10 water, methanol, and acetonitrile (*v*/*v*). The column temperature was 40 °C. The pump flow rate was 0.8 mL/min.

To prepare the HPLC standard curve, a certified reference material aflatoxin mix was obtained from Sigma-Aldrich (CRM46304; St. Louis, MO, USA). The aflatoxin mix contained 1000 µg/kg AFB1, 1000 µg/kg AFG1, 300 µg/kg AFB2, and 300 µg/kg AFB2. The standards were calibrated on the HPLC using the aflatoxin mix detection established by the column manufacturer [27] with a higher injection volume (20 µL instead of 10 µL) to ensure detection. LOD was 5 ppb using a minimum signal-to-noise ratio of 2:1, and LOQ was 20 ppb using a signal-to-noise ratio of 10:1 combined with recovery calculations ± 20% (0.8 to 1.2; Appendix A). A seven-point standard curve (5, 10, 20, 50, 100, 200, and 500 ppb) with triplicate readings had an R^2^ value of 0.9989.

### 4.7. Aspergillus flavus Quantification

DNA was extracted from 250 mg of each irradiated sample using a Qiagen PowerSoil kit (Germantown, MD, USA) with the modification of a 50 µL final elution. The extracted DNA was stored at −80 °C. Moreover, 8 μL of DNA was used for qPCR with a Youseq *A. flavus* qPCR Test Kit (CAT No.: YSL-qP-IC-A.flavu-100, Youseq Ltd., Winchester, UK), which targets the MDR1 gene and uses Taqman chemistry. PCR was performed with an Applied Biosystems QuantStudio 3 Real-Time PCR (Thermo Fisher Scientific Corp., Waltham, MA, USA).

Colony-forming units (CFU) were determined after suspending 2 g of each sample in 8 mL of sterile 0.1% Triton water and mixing on a rotary shaker for 30 min. Ten-fold serial dilutions were performed to a final dilution of 10^−6^ and 100 uL aliquots from each dilution spread on MDRB and PDA plates with a flame-sterilized spreader. Plates were sealed with parafilm and stored in an incubator at 37 °C for 3 days. Colony-forming units were counted for each plate at 66 h.

### 4.8. Statistics

Descriptive statistics and a one-way ANOVA test for statistical differences were performed using R Studio Software (version 2023.06.2 + 561). Significance is noted by *p* < 0.05, with additional notations for significance (NS. for *p* ≥ 0.05, * for *p* < 0.05, ** for *p* < 0.01, *** for *p* < 0.001).

## Figures and Tables

**Figure 1 toxins-16-00329-f001:**
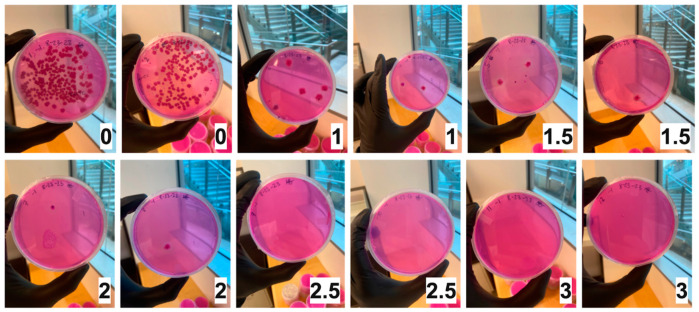
Duplicate irradiated maize samples plated on modified Rose Bengal agar (MDRB) in serial dilutions. A one-hundred-fold dilution is shown. Top left to bottom right images display increasing irradiation doses from 0.0 kilogray (kGy) to 3.0 kGy, indicated in the bottom right of each image by a black number in a white box.

**Figure 2 toxins-16-00329-f002:**
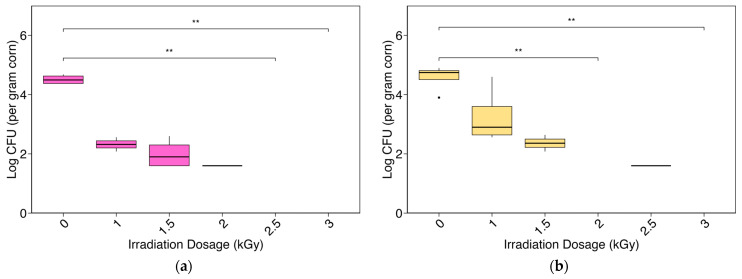
Box plots represent serial dilution plating (tenfold) of X-ray-irradiated maize samples: (**a**) modified Rose Bengal agar (MDRB); (**b**) potato dextrose agar (PDA). All plates with observed samples are reported. Significant results between treatments are marked on plots (** for *p* < 0.01).

**Figure 3 toxins-16-00329-f003:**
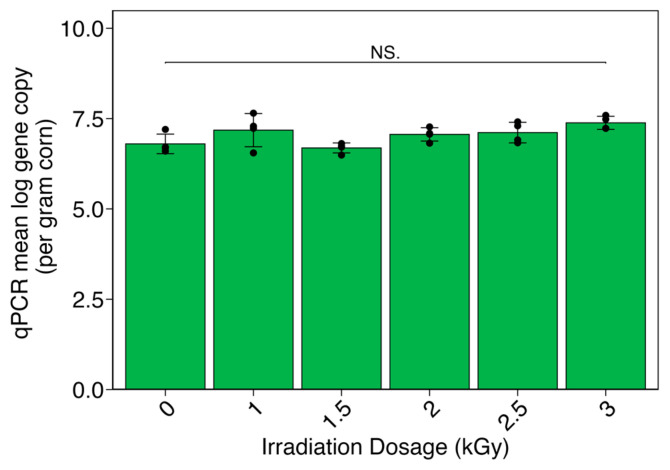
Quantitative polymerase chain reaction (qPCR) results for the MDR1 gene in X-ray-irradiated maize samples. Data points represent logarithmic values of duplicate qPCR measurements for duplicate samples. Error bars represent the mean ± the standard deviation. No significant differences were noted between any of the irradiation doses (NS. for *p* ≥ 0.05).

**Figure 4 toxins-16-00329-f004:**
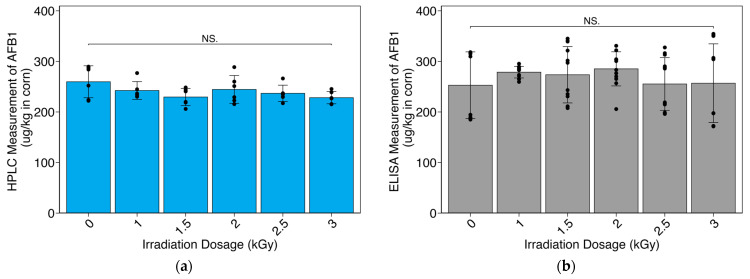
Aflatoxin B1 (AFB1) quantification in X-ray-irradiated samples using high-performance liquid chromatography (HPLC) and an enzyme-linked immunosorbent assay (ELISA): (**a**) HPLC data represent triplicate readings of duplicate samples; (**b**) ELISA data represent triplicate readings of duplicate samples that were diluted twice at a 1:1 ratio to ensure samples were detected by kit detection limits (2–50 ppb). AFB1 concentration is reported in µg/kg (parts per billion). Error bars represent the mean ± the standard deviation. No significant differences are noted between any of the irradiation doses for HPLC or ELISA (NS. for *p* ≥ 0.05).

**Figure 5 toxins-16-00329-f005:**
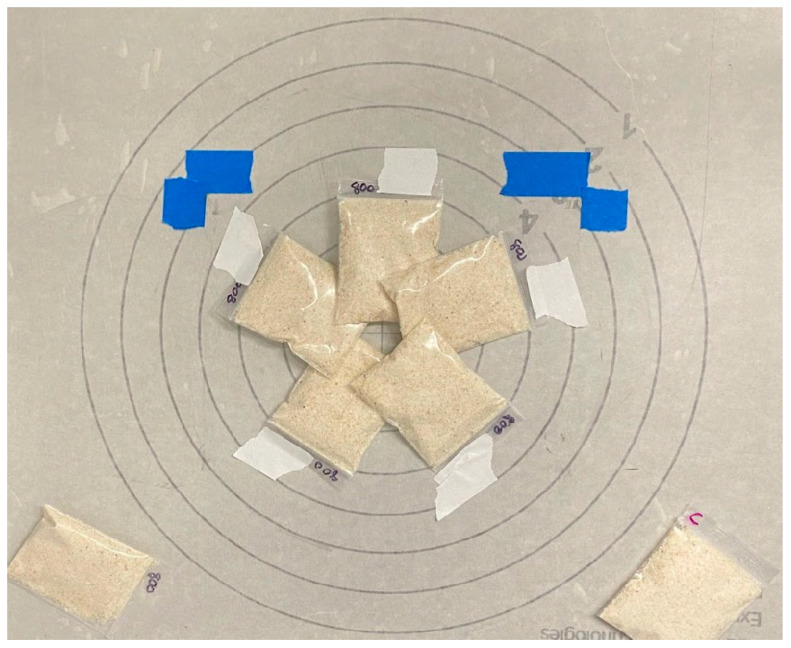
Alignment of the bags of maize at level 5 on the X-ray irradiator plate after the no-dosage bag was removed. Control bags of corn for AFB1 contamination and non-contamination were kept on the plate outside of the zone of irradiation.

## Data Availability

The original contributions presented in the study are included in the article and Appendix A. Further inquiries can be directed to the corresponding authors. The raw data supporting the conclusions of this article will be made available by the authors on request.

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
