# Peer review of "X-ray Irradiation Reduces Live Aspergillus flavus Viability but Not Aflatoxin B1 in Naturally Contaminated Maize"

_toxins, 2024, doi:10.3390/toxins16080329_

Round 1

Reviewer 1 Report

Comments and Suggestions for Authors

The manuscript ID toxins-308314 entitled " X-ray Irradiation reduces live Aspergillus flavus but not Aflatoxin B1 in Contaminated Maize", is an interesting study. But my suggest it needs to a major revision to be published in this journal. Some specific comments:

(1) The authors claimed that “It is unknown whether the X-ray irradiation results seen in cannabis flower will translate to contaminated maize, though promising results from the gamma irradiation studies point to this likelihood”, what is the difference, just change a host or material? The originality should be described clearly.

(2)     Abstract: Revise and provides a clear overview of the study, novelty and its findings.

(3)     Materials and Methods:

What is the source of Aspergillus flavus?

the detailed information of the reagents should be described (brand, city, country)

(4)     Discussion: It was written in a scientifically correct manner.

(5)  Conclusion:

There is no conclusion

(6)      Reference: The references are updated and written correctly.

Author Response

Dear Reviewer,

We are very grateful for your time and energy in reviewing our manuscript and the
suggestions you have provided. We have attempted to fully integrate them into the
manuscript and believe it is strong because of it. Please let us know if you have
further suggestions or thoughts on how we can improve it.

Sincerely,
The Authors

Reviewer 1        

The manuscript ID toxins-308314 entitled " X-ray Irradiation reduces live Aspergillus flavus but not Aflatoxin B1 in Contaminated Maize", is an interesting study. But my suggest it needs to a major revision to be published in this journal. Some specific comments:

Comment 1: The authors claimed that “It is unknown whether the X-ray irradiation results seen in cannabis flower will translate to contaminated maize, though promising results from the gamma irradiation studies point to this likelihood”, what is the difference, just change a host or material? The originality should be described clearly.

Response 1: Thank you for the opportunity to provide an improved explanation. We have revised the selected sentence for clarity. “It is unknown whether the X-ray irradiation results seen in cannabis flower will translate to contaminated maize because of the different material matrix, though A. flavus reduction in maize gamma irradiation studies point to the likelihood that x-ray irradiation will also reduce A. flavus viability in maize.” Changes can be seen on page 2, paragraph 4, lines 100-104.

Comment 2: Abstract: Revise and provides a clear overview of the study, novelty and its findings.

Response 2: Thank you for the opportunity to provide a clear abstract. We have made several updates to clearly state the study, novelty and findings. For example, we updated lines 11-12 to read: “Maize that had been naturally contaminated with 300 ppb AFB1 by growth of aflatoxigenic A. flavus was ground and then irradiated at 0.0, 1.0, 1.5, 2.0, 2.5 and 3.0 kGy.”

Comment 3: Materials and Methods: What is the source of Aspergillus flavus? the detailed information of the reagents should be described (brand, city, country)

Response 3: We regret this omission.  The text at line 261 has been revised to include the relevant information and now reads: “..this maize was moistened and inoculated in a biosafety cabinet with a mix of spores of aflatoxigenic A. flavus collected from Guatemala, and representing common genotypes, as verified by DNA microsatellite analysis [2].  The maize was incubated, with samples collected and analyzed for AFB1 every 8 – 12 hours until the target concentration of 300 ppb AFB1 was reached.” Changes can be seen on page 7, paragraph 2, lines 260-264.

In addition, thank you for the suggestion to describe all reagents mentioned. We updated the reagents section to include all the chemicals used in the rose bengal media. We also edited this section for grammar (removing capitalization of chemical names). Changes can be seen on page 7, paragraph 1, lines 237-247.

Comment 4: Discussion: It was written in a scientifically correct manner.

Response 4: No response necessary.  

Comment 5: Conclusion: There is no conclusion

Response 5: No response necessary.  

Comment 6: Reference: The references are updated and written correctly.

Response 6: No response necessary.  

Reviewer 2 Report

Comments and Suggestions for Authors

I have read the paper “X-ray Irradiation reduces live Aspergillus flavus but not Aflatoxin B1 in Contaminated Maize.” This work is interesting. It provides a strategy for obtaining natural contamination samples with the specific AFB1 level and reduced live Aspergillus flavus. These natural contamination samples will be more helpful in studying the AFB1 translation or elimination than artificially spiked ones. However, the method used in this paper for the quantitative analysis of AFB1 needs further consideration.

Specific comments

1. According to COMMISSION REGULATION (EC) No 401/2006, if ELISA is used as a screening method, the positive results of samples should be confirmed by a confirmatory method such as HPLC or LC-MS/MS. An accurate result could be obtained by HPLC; thus, it seems unnecessary to use ELISA for detection in this work.  

2. Due to its limited sensitivity, the PDA detector is rarely used when determining AFB1 by HPLC. What is the LOD of the HPLC method in this work? If the method validation was performed? These conditions should be indicated.

3. For ELISA, considering the tolerance of antibodies to organic solvents, the concentration of organic solvent in the sample solution after dilution is usually not more than 20%; however, in this work, the concentration of MeOH is 70% in the diluted sample solution, which seems inconsiderable and should be confirmed.

4. Lines 246-248, the author should indicate the necessity of the two dilutions. The total dilution factor is 8, so why single dilution was not considered? In addition, in line 249, “two diluted samples (1:4, 1:9)”, 1:9 means?  

5. Line 41, the format of reference 5 should be revised. 

Author Response

Dear Reviewer,

We are very grateful for your time and energy in reviewing our manuscript and the
suggestions you have provided. We have attempted to fully integrate them into the
manuscript and believe it is strong because of it. Please let us know if you have
further suggestions or thoughts on how we can improve it.

Sincerely,
The Authors

Reviewer 2

I have read the paper “X-ray Irradiation reduces live Aspergillus flavus but not Aflatoxin B1 in Contaminated Maize.” This work is interesting. It provides a strategy for obtaining natural contamination samples with the specific AFB1 level and reduced live Aspergillus flavus. These natural contamination samples will be more helpful in studying the AFB1 translation or elimination than artificially spiked ones. However, the method used in this paper for the quantitative analysis of AFB1 needs further consideration.

Specific comments

Comment 1: According to COMMISSION REGULATION (EC) No 401/2006, if ELISA is used as a screening method, the positive results of samples should be confirmed by a confirmatory method such as HPLC or LC-MS/MS. An accurate result could be obtained by HPLC; thus, it seems unnecessary to use ELISA for detection in this work.  

Response 1: We agree with your comment that a positive result for ELISA should be confirmed by HPLC when officially testing food samples that will be consumed. However, the decision to include ELISA in this study was made because it is often much more accessible and affordable for researchers and regional regulatory stakeholders who may not have an HPLC and would like to test AFB1 concentrations.

In addition, we believe it is important to include ELISA as a method because it continues to provide additional data around the specific capabilities of these types of kits that may be a more feasible option in lower resource settings. These settings are often the places where AFB1 is found in higher and more frequent concentrations.

Comment 2: Due to its limited sensitivity, the PDA detector is rarely used when determining AFB1 by HPLC. What is the LOD of the HPLC method in this work? If the method validation was performed? These conditions should be indicated.

Response 2: Thank you for the opportunity to clarify our HPLC methodology. We recognize that PDA detectors are rarely used today when determining AFB1 by HPLC, however the levels of AFB1 we were working with were sufficient to be detected using previously established UV detection methods (Barbas; now reference 25 in the manuscript). In addition, FLD are less common than PDA detectors and we think it is important to explore alternative detectors that are more widely available. Our lab did not have an FLD and given the high cost of purchasing a detector, we made the choice to test a method using the in-house PDA detector. This segways to your second line of inquiry about the HPLC method validation and LOD.

We regret the omission of these details and have heavily revised the HPLC methods section to include this information: “To prepare the HPLC standard curve, a certified reference material aflatoxin mix was obtained from Sigma-Aldrich (CRM46304; St. Louis, MO, USA). The aflatoxin mix contained 1000 µg/kg AFB1, 1000 µg/kg AFG1, 300 µg/kg AFB2, and 300 µg/kg AFB2. The standards were calibrated on the HPLC using the aflatoxin mix detection established by the column manufacturer [25] with a higher injection volume (20 µl instead of 10 µl) to ensure detection. LOD was 5 ppb using a minimum signal to noise ratio of 2 to 1 and LOQ was 20ppb using a signal to noise ratio of 10 to 1 combined with recovery calculations ± 20% (0.8 to 1.2; Supplementary Table 8 and Supplementary Figure 1). A seven-point standard curve (5, 10, 20, 50, 100, 200, and 500 ppb) with triplicate readings had an R2 value of 0.9989.” Changes can be found on page 9, paragraph 3, lines 392-401.

Ref: Barbas, C., Dams, A., & Majors, R. E. (2005). Analysis of organic acids using high-performance liquid chromatography. Agilent Technologies. Retrieved from https://www.agilent.com/Library/applications/5989-3634EN.pdf”

Comment 3: For ELISA, considering the tolerance of antibodies to organic solvents, the concentration of organic solvent in the sample solution after dilution is usually not more than 20%; however, in this work, the concentration of MeOH is 70% in the diluted sample solution, which seems inconsiderable and should be confirmed

Response 3: Thank you for the opportunity to clarify the ELISA methodology. We thought it was important to dilute our samples with 70% methanol as the Romer Agraquant ELISA kit we selected was designed to handle 70% methanol extractions. The kit included an additional dilution step with a proprietary buffer, which likely diluted the extract to a tolerable solvent concentration for the antibody assay. In addition, it is likely the kit manufactures chose methanol not only because it is widely available and used in AFB1 extractions, but also because methanol is thought to interfere less than other organic solvents (e.g. acetonitrile) with AFB1 immunoassay antibodies (Bertuzzi et al., 2012 – reference 12 in manuscript). Thank you again for the thoughtful question and attention to detail.

Bertuzzi, T.; Rastelli, S.; Mulazzi, A.; Pietri, A. Evaluation and Improvement of Extraction Methods for the Analysis of Aflatoxins B1, B2, G1 and G2 from Naturally Contaminated Maize. Food Anal. Methods 2012, 5 (3), 512–519. https://doi.org/10.1007/s12161-011-9274-5.

Comment 4: Lines 246-248, the author should indicate the necessity of the two dilutions. The total dilution factor is 8, so why single dilution was not considered? In addition, in line 249, “two diluted samples (1:4, 1:9)”, 1:9 means?  

Response 4: This question is important. We heavily revised the mentioned lines to clarify why two dilutions were necessary and what the ratios mean, as well as providing a thorough explanation of method parameters: “The kit range of quantification was 2-50 ppb with a limit of detection (LOD) and limit of quantification (LOQ) of 2 ppb. Each maize filtrate was diluted twice to ensure AFB1 quantification given the small range of AFB1 detection. Briefly, 10 mL of each sample filtrate was diluted with 40 mL of 70% methanol. Half of the resulting filtrated (25 mL) was further diluted with a 1:1 ratio of 25 mL 70% methanol (1:9 of original filtrate). The two diluted sample filtrates (100 uL of 1:4 and 1:9) were quantified for AFB1 following the AgraQuant kit instructions.” Changes can be found on page 8, final paragraph, lines 362-381.

In addition, the caption of Figure 4 b has been updated to reflect the methods description: “(b) ELISA data represents triplicate readings of duplicate samples that were diluted twice at a 1:1 ratio to ensure samples were detected by kit detection limits (2-50 ppb).” Changes can be found on page 5, lines 172-173

Comment 5: Line 41, the format of reference 5 should be revised. 

Response 5: Thank you for revealing this error. The formatting has been updated.

Reviewer 3 Report

Comments and Suggestions for Authors

The manuscript titled "X-ray Irradiation reduces live Aspergillus flavus but not Aflatoxin B1 in Contaminated Maize" is describing a research and results that is testing x-ray irradiation as a method to deactivate toxin Aspergillus flavus fungi in maize without altering the concentration of aflatoxin B1 mycotoxin that is produced by the fungi. Authors suggests various method are available to use either chemical or physical to deactivate the fungi. These processes have limitations and the suggested x-ray irradiation has better applicability for research activities. The authors are aiming to establish this method as choice of sterilization in laboratory research settings.

The authors also discuss some of the potential limitations of this method.

The manuscript is fairly well written, easy to understand and synthesize the information, results and discussion.

There are minor comments/suggestions to further improve the manuscript.

Lines 9-11: Revise objective. the current statement is unclear. establish a sterilization method - instead use testing a sterilization method... sterilization method for studying AFB1?? sterilization method to study the effect on AFB1 using X-ray irradiation?

Line 11-12: how was A. flavus targeted to 300 ppb? is it the concentration of toxins or the fungi? If it is fungi, how was 300 ppb acheived and how it is relevant? If it is toxin, was it spiked or let the fungi accumulate the toxin? Revise sencente for clarity. While it is clear in the method section, it doesn't have similar description in the abstract. The relevant text in the abstract is unclear.

Lines 175-180, 216-217, 220-221: While the main objective of the study was to establish a x-ray irridiation method to denature toxic fungi prior to using them in research studies, however, a significant handling of contaminated grain is required to prepare the sample for x-ray irridiation which doesn't serve the purpose. The grinding of the maize grain with toxic fungi to prepare for x-ray is the step being refered in this comment. While this is limitation as appropirately included in the discussion, just wanted to note that the main objective is not fully achieved. This comment is 'food for thought'. May be testing x-rays on contaminated grain with minimal handling prior to x-rays would be of great addition to this study.

Line 249, 257, 258: avoid starting sentence with a number or abbreviation.

Line 212-214: The maize was heated at 90C for 30 min: how does this effect the A. flavus. Does this process have any growth enhancement or suppression effect. Assuming the same grain was used for control and treatments and effect would have been normalized, it would be important to identify the effect and isolate that the effects observed were directly from irradiation and not a combination.

It was quite noticeable that the authors didn't mention positive control and negative controls in the methods, results, nor they were discussed. They were only present in the supplementary information. Suggest to use that in results and discuss.

The concentration of AFB1 in positive control was <LOD. Explain. The concentration of AFB1 in negative control was 205.26 and 190.54. Explain. Provide details as what/how were positive and negative controls used that could explain the presence/absence of AFB1.

Lines 243-260: Citation for HPLC method used for AFB1 extraction and quantification. If the method was developed, validation parameters need to be included.

Author Response

Dear Reviewer,

We are very grateful for your time and energy in reviewing our manuscript and the
suggestions you have provided. We have attempted to fully integrate them into the
manuscript and believe it is strong because of it. Please let us know if you have
further suggestions or thoughts on how we can improve it.

Sincerely,
The Authors

Reviewer 3

The manuscript titled "X-ray Irradiation reduces live Aspergillus flavus but not Aflatoxin B1 in Contaminated Maize" is describing a research and results that is testing x-ray irradiation as a method to deactivate toxin Aspergillus flavus fungi in maize without altering the concentration of aflatoxin B1 mycotoxin that is produced by the fungi. Authors suggests various method are available to use either chemical or physical to deactivate the fungi. These processes have limitations and the suggested x-ray irradiation has better applicability for research activities. The authors are aiming to establish this method as choice of sterilization in laboratory research settings.

The authors also discuss some of the potential limitations of this method.

The manuscript is fairly well written, easy to understand and synthesize the information, results and discussion.

There are minor comments/suggestions to further improve the manuscript.

Comment 1: Lines 9-11: Revise objective. the current statement is unclear. establish a sterilization method - instead use testing a sterilization method... sterilization method for studying AFB1?? sterilization method to study the effect on AFB1 using X-ray irradiation?

Response 1: We agree that the wording could be revised. Thank you. Lines 9-11 now read, “The objective of this study is to test an X-ray irradiation sterilization method for studying AFB1 in naturally contaminated maize sample in the laboratory.”

Comment 2: Line 11-12: how was A. flavus targeted to 300 ppb? is it the concentration of toxins or the fungi? If it is fungi, how was 300 ppb acheived and how it is relevant? If it is toxin, was it spiked or let the fungi accumulate the toxin? Revise sencente for clarity. While it is clear in the method section, it doesn't have similar description in the abstract. The relevant text in the abstract is unclear.

Response 2: Thank you for the opportunity to clarify this point.  We have added detail in the methods (see additional text at line 264) and made a change at line 11-12 to read: “Maize that had been naturally contaminated with 300 ppb AFB1 by growth of aflatoxigenic A. flavus was ground and then irradiated at 0.0, 1.0, 1.5, 2.0, 2.5 and 3.0 kGy.”

Comment 3: Lines 175-180, 216-217, 220-221: While the main objective of the study was to establish a x-ray irridiation method to denature toxic fungi prior to using them in research studies, however, a significant handling of contaminated grain is required to prepare the sample for x-ray irridiation which doesn't serve the purpose. The grinding of the maize grain with toxic fungi to prepare for x-ray is the step being refered in this comment. While this is limitation as appropirately included in the discussion, just wanted to note that the main objective is not fully achieved. This comment is 'food for thought'. May be testing x-rays on contaminated grain with minimal handling prior to x-rays would be of great addition to this study.

Response 3: Thank you for the insightful comment. This is an important ‘food for thought’ comment. We agree with your general assessment. The key factor will be the source of the contaminated maize—where the lab interested in studying maize acquires it from (direct from field, grown by a different lab, inoculated themselves, etc.). This could vary widely. In our scenario, there was a lab that specialized in A. flavus and AFB1 research. They grew the maize, ground it, and sent it to us. Our trade-off decision was if they should have ground the maize or left it whole kernel. The reason we chose an initial grinding step prior to acquiring it was that this significantly improved the homogenization of AFB1 in the sample, which was important for our smaller sample mass and allowed us to reduce a potential significant amount of noise in the data. But we lost the ability to answer a part of the question about irradiation penetration of the kernel. However, the actual handling of contaminated maize pre-X-ray remained the same. We received the shipment in ‘ready-to X-ray’ material from the specialized lab and immediately subjected the samples to X-ray treatment upon receipt. This would be the same handling procedure if it were whole kernel. In the future, testing X-rays with contaminated grain with whole kernel will be helpful to verify effectiveness, as you pointed out and we relay in the discussion.

Comment 4: Line 249, 257, 258: avoid starting sentence with a number or abbreviation.

Response 4: Thank you for pointing these errors out. We re-worded sentences you pointed out that started with a number or abbreviation. Changes can be found on page 8, paragraph 2, lines 353-356 and on page 8, final paragraph, lines 360-366.

Comment 5: Line 212-214: The maize was heated at 90C for 30 min: how does this effect the A. flavus. Does this process have any growth enhancement or suppression effect. Assuming the same grain was used for control and treatments and effect would have been normalized, it would be important to identify the effect and isolate that the effects observed were directly from irradiation and not a combination.

Response 5: The reviewer is correct that this heat treatment may have altered the microflora.  In preliminary experiments we observed that incubating moistened maize after this heat treatment there was a noticeable reduction in nuisance microflora.  A. flavus is very heat-tolerant and while this treatment likely killed some A. flavus, some would survive such treatment.  The reviewer is correct that the maize for all treatments received the same 90C exposure.  Nearly all of the A. flavus exposed to the irradiation was A. flavus that grew from the inoculation and incubation, which occurred after the heat exposure, so the radiation-induced mortality is independent of the heat exposure.   

Comment 6: It was quite noticeable that the authors didn't mention positive control and negative controls in the methods, results, nor they were discussed. They were only present in the supplementary information. Suggest to use that in results and discuss.

Response 6: We agree that the experimental controls should be described and regret this omission. Details on the controls used have been included in the methods section. The negative control was produced in the maize inoculation step and the text has been revised to include this information: “A portion of the grain sample was aliquoted as a negative control for the X-ray irradiation experiment, while the rest of this maize was moistened and inoculated in a biosafety cabinet with a mix of spores of aflatoxigenic A. flavus from Guatemala, and representing common genotypes, as verified by DNA microsatellite analysis [2].” Changes can be found on page 7, paragraph 2, lines 260-264.

The negative and positive controls were processed during the sample aliquot step. The text has been updated to include the control information: “Experimental controls were also aliquoted in the biosafety cabinet in the same portions as experimental samples. The negative control was a portion of the composite grain sample before A. flavus contamination. The positive control was a certified reference material with a known AFB1 concentration purchased from Trilogy Analytical Laboratory (Washington, MO, USA).” Changes can be found on page 8, paragraph 1, lines 344-348.

Comment 7: The concentration of AFB1 in positive control was <LOD. Explain. The concentration of AFB1 in negative control was 205.26 and 190.54. Explain. Provide details as what/how were positive and negative controls used that could explain the presence/absence of AFB1.

Response 7: Thank you for pointing out this error and providing the opportunity to correct it. Supplemental Table 3 includes the data you mentioned, where the negative control displayed 205.26 ppb AFB1 in the ELISA assay and 190.54 ppb AFB1 in the HPLC assay, while the positive control was <LOD. Regretful, we populated this dashboard in the incorrect cell. We have corrected this table to reflect the results we compiled in our raw data, which show the negative control was in fact <LOD and the positive control was 205.26 ppb AFB1 in the ELISA assay and 190.54 ppb AFB1 in the HPLC assay. The updated Supplemental Table 3 displays the correct results. Thank you again for the attention to detail.

Comment 8: Lines 243-260: Citation for HPLC method used for AFB1 extraction and quantification. If the method was developed, validation parameters need to be included.

Response 8: We strongly agree the HPLC method used for AFB1 citation should be included and regret this oversight. We heavily revised the HPLC methods section to include this information as well as additional analytical methodology: “To prepare the HPLC standard curve, a certified reference material aflatoxin mix was obtained from Sigma-Aldrich (CRM46304; St. Louis, MO, USA). The aflatoxin mix contained 1000 µg/kg AFB1, 1000 µg/kg AFG1, 300 µg/kg AFB2, and 300 µg/kg AFB2. The standards were calibrated on the HPLC using the aflatoxin mix detection established by the column manufacturer [25] with a higher injection volume (20 µl instead of 10 µl) to ensure detection. LOD was 5 ppb using a minimum signal to noise ratio of 2 to 1 and LOQ was 20ppb using a signal to noise ratio of 10 to 1 combined with recovery calculations ± 20% (0.8 to 1.2; Supplementary Table 8 and Supplementary Figure 1). A seven-point standard curve (5, 10, 20, 50, 100, 200, and 500 ppb) with triplicate readings had an R2 value of 0.9989.” Changes can be found on page 9, paragraph 3, lines 393-402.

Reviewer 4 Report

Comments and Suggestions for Authors

The manuscript entitled “X-ray irradiation reduces live Aspergillus flavus but not aflatoxin B1 in contaminated maize” presented a study on X-ray irradiation on naturally mycotoxin contaminated maize sample, to evaluate the effects after ionizing irradiation treatments, analyzing aflatoxin B1 contents and A. flavus viability. X-ray irradiation at 3.0 kGy reduced viable A. flavus without affect aflatoxin B1 initial concentrations. Overall, the manuscript seems to be well structured and written. While the topic of the work is interesting, the results obtained may be better discussed. I suggest to improve the discussion part by comparing the results with previous studies using other ionizing radiation methods. The relevance of the obtained results was increased by contrasting a well established analytical technique such as HPLC with ELISA method for analyzing mycotoxin concentrations. However, some modifications are need before it is accepted for publication.

  1. I suggest to complete the title with: “X-ray irradiation reduces Aspergillus flavus viability but not Aflatoxin B1 in naturally contaminated maize”.
  2. L 10: change “…in contaminated food products…” with “…in contaminated maize sample”.
  3. L 29 and 41: to change reference citation method: “…due to crop loss1” with “…due to crop loss [1]”.
  4. L 68: The authors should include additional information about X-ray irradiation at Introduction section, discussing the significance in the context of the food industry and highlight potential applications and benefits.
  5. L 113, 125, 136: to change “(kilogray)” with “(kGy)”, at Figures 2, 3 and 4.
  6. L 128: to change “Error bars represent mean plus or minus the standard deviation” with “Error bars represent mean ± standard deviation”.
  7. L 140: to change “ug/kg” with “µg/kg”.
  8. L 141: to change “Error bars represent mean plus or minus the standard deviation” with “Error bars represent mean ± standard deviation”.
  9. L 172: “FDA” this acronym should be define in the text, the first time.
  10. L 200: Complete chemical information of AFB1 mycotoxin standard should be provided.
  11. L 202: The authors could check the entire manuscript to insert a space between the numeric value from the unit: at time, temperature data and numerical values, v.g. “4ºC”, the correct form is “4 ºC”.
  12. L 215: At this point, data regarding fungal isolation and identification method should be provided in the text.
  13. L 213: change “minutes” with “min” and revise throughout the text.
  14. L 243: The authors should include LOD data of the analytical methods used. Also, the R2 coefficient between both methods could be valuable.

Author Response

Dear Reviewer,

We are very grateful for your time and energy in reviewing our manuscript and the
suggestions you have provided. We have attempted to fully integrate them into the
manuscript and believe it is strong because of it. Please let us know if you have
further suggestions or thoughts on how we can improve it.

Sincerely,
The Authors

Reviewer 4

The manuscript entitled “X-ray irradiation reduces live Aspergillus flavus but not aflatoxin B1 in contaminated maize” presented a study on X-ray irradiation on naturally mycotoxin contaminated maize sample, to evaluate the effects after ionizing irradiation treatments, analyzing aflatoxin B1 contents and A. flavus viability. X-ray irradiation at 3.0 kGy reduced viable A. flavus without affect aflatoxin B1 initial concentrations. Overall, the manuscript seems to be well structured and written. While the topic of the work is interesting, the results obtained may be better discussed. I suggest to improve the discussion part by comparing the results with previous studies using other ionizing radiation methods. The relevance of the obtained results was increased by contrasting a well established analytical technique such as HPLC with ELISA method for analyzing mycotoxin concentrations. However, some modifications are need before it is accepted for publication.

Comment 1: I suggest to complete the title with: “X-ray irradiation reduces Aspergillus flavus viability but not Aflatoxin B1 in naturally contaminated maize”.

Response 1: Thank you for this title suggestion. We appreciate the clarity that this wording provides and have adjusted our manuscript title to include “viability” and “naturally”.

Comment 2: L 10: change “…in contaminated food products…” with “…in contaminated maize sample”.

Response 2: Thank you for this suggestion. We revised the entire sentence, and it now reads “The objective of this study is to test an X-ray irradiation sterilization method for studying AFB1 in contaminated maize sample in the laboratory”. Changes can be found on page 1, paragraph 1, lines 9-11.

Comment 3: L 29 and 41: to change reference citation method: “…due to crop loss1” with “…due to crop loss [1]”.

Response 3: Thank you for pointing this out. The formatting has been changed from superscript to bracket. 

Comment 4: L 68: The authors should include additional information about X-ray irradiation at Introduction section, discussing the significance in the context of the food industry and highlight potential applications and benefits.

Response 4: We appreciate the suggestion to include additional information about X-ray irradiation in the introduction to provide context for the significance of this experiment. We have updated the introduction to include this context: “The third option is X-ray irradiation, which is estimated to be widely used in the future for food safety [17]. X-ray irradiation can be adapted at smaller scale without complex operating facilities, which means that units can be directly incorporated in continuous local food processes, saving the transportation and handling costs when a larger, remote facility is used for irradiation [4]. Companies like Rayfresh Foods are working towards building this technology, though adoption is a hurdle due to cost of implementation and maintenance. Electricity costs are also high, which is a downside to X-ray irradiation yet the market for smaller, safer, and easily operated irradiation technology is increasing at the same time as research into improved efficiency of renewable electricity [8]. Therefore, we chose X-ray irradiation as the cold-pasteurization method for this study.

Comment 5: L 113, 125, 136: to change “(kilogray)” with “(kGy)”, at Figures 2, 3 and 4.

Response 5: We appreciate the attention to detail and agree the figure text should be changed. Figures 2, 3, and 4 have been updated to display “(kGy)” instead of “(kilogray)”.  

Comment 6: L 128: to change “Error bars represent mean plus or minus the standard deviation” with “Error bars represent mean ± standard deviation”.

Response 6: Thank you. Line 156 has been updated with “±” instead of “plus or minus”.

Comment 7: L 140: to change “ug/kg” with “µg/kg”.

Response 7: Thank you for pointing this out. The formatting has been changed from “u” to “µ”.

Comment 8: L 141: to change “Error bars represent mean plus or minus the standard deviation” with “Error bars represent mean ± standard deviation”.

Response 8: Thank you. Line 173 has been updated with “±” instead of “plus or minus”.

Comment 9: L 172: “FDA” this acronym should be define in the text, the first time.

Response 9: Thank you for pointing this out. The acronym at line 212 has been spelled out as “the U.S. Food and Drug Administration (FDA)”.

Comment 10: L 200: Complete chemical information of AFB1 mycotoxin standard should be provided.

Response 10: Thank you for the opportunity to provide more information about the AFB1 standard. We have updated the sentence including the AFB1 mycotoxin standard to include the certified reference material identifier and AFB1 concentration. “All the following chemicals were purchased from Sigma-Aldrich (St. Louis, MO, USA): methanol (≥99.9%), acetonitrile (≥99.9%), acetone (≥99.5%), sodium chloride (≥99.0%, CAS: 7647-14-5), peptone (CAS: 91079-38-8), yeast extract (CAS: 8013-01-2), magnesium sulfate heptahydrate (≥99.0%, CAS: 10034-99-8), potassium phosphate monobasic (≥99.0%, CAS: 7778-77-0), rose bengal (≥80%, CAS: 632-69-9), streptomycin sulfate salt (CAS: 3810-74-0), chlortetracycline hydrochloride (≥75%, CAS: 64-72-2), 2, 6-dichloro-4-nitroaniline (96%, CAS: 99-30-9), agar powder (CAS: 9002-18-0), triton-X 100, potato dextrose agar (PDA) and aflatoxin mix (certified reference material: CRM46304, 1000 ug/kg AFB1).” Changes can be found on page 7, paragraph 1, lines 237-245.

Comment 11: L 202: The authors could check the entire manuscript to insert a space between the numeric value from the unit: at time, temperature data and numerical values, v.g. “4ºC”, the correct form is “4 ºC”.

Response 11: Thank you for pointing this out. We inserted a space between the numeric values of all units: time, temperature data and numerical values.

Comment 12: L 215: At this point, data regarding fungal isolation and identification method should be provided in the text.

Response 12: We regret this omission.  The text at line 261 has been revised to include the relevant information and now reads: “This maize was moistened and inoculated in a biosafety cabinet with a mix of spores of aflatoxigenic A. flavus collected from Guatemala, and representing common genotypes, as verified by DNA microsatellite analysis [2].  The maize was incubated, with samples collected and analyzed for AFB1 every 8 – 12 hours until the target concentration of 300 ppb AFB1 was reached.” Changes can be found on page 7, paragraph 2, lines 260-264.

Comment 13: L 213: change “minutes” with “min” and revise throughout the text.

Response 13: Thank you for pointing this out. We revised the text to replace all usage of “minutes” to “min”.

Comment 14: L 243: The authors should include LOD data of the analytical methods used. Also, the R2 coefficient between both methods could be valuable.

Response 14: We strongly agree the LOD data of the analytical methods should be included and regret this oversight. We also incorporated the suggestion to include R2 values.

For the ELISA analytical method, we adjusted the section to include: “The kit range of quantification was 2-50 ppb with an limit of detection (LOD) and limit of quantification (LOQ) of 2 ppb. Each maize filtrate was diluted twice to ensure AFB1 quantification given the small range of AFB1 detection. Briefly, 10 mL of each sample filtrate was diluted with 40 mL of 70% methanol. Half of the resulting filtrated (25 mL) was further diluted with a 1:1 ratio of 25 mL 70% methanol (1:9 of original filtrate). The two diluted sample filtrates (100 uL of 1:4 and 1:9) were quantified for AFB1 following the AgraQuant kit instructions. ELISA 96 well plates were read on a BioTek Synergy HTX Multimode Reader (Agilent Technologies, Santa Clara, CA, USA) with the monochromator set at 450 nm. Using kit provided AFB1 solutions, a five-point standard curve (0, 2, 5, 20, and 50 ppb) with triplicate readings had an R2 value of 0.9908.” Changes can be found on page 8, final paragraph, lines 362-384.

For the HPLC analytical method, we adjusted the section to include: “LOD was 5 ppb using a minimum signal to noise ratio of 2 to 1 and LOQ was 20ppb using a signal to noise ratio of 10 to 1 combined with recovery calculations ± 20% (0.8 to 1.2; Supplementary Table 8 and Supplementary Figure 1). A seven-point standard curve (5, 10, 20, 50, 100, 200, and 500 ppb) with triplicate readings had an R2 value of 0.9989.” Changes can be found on page 9, paragraph 3, lines 398-402.

Round 2

Reviewer 1 Report

Comments and Suggestions for Authors

The authors made sufficient modification, it can be accepted.

Reviewer 2 Report

Comments and Suggestions for Authors

The paper has been revised and improved.